# Application of Machine Learning Methods to Analyze Occurrence and Clinical Features of Ascending Aortic Dilatation in Patients with and without Bicuspid Aortic Valve

**DOI:** 10.3390/jpm12050794

**Published:** 2022-05-14

**Authors:** Olga Irtyuga, Georgy Kopanitsa, Anna Kostareva, Oleg Metsker, Vladimir Uspensky, Gordeev Mikhail, Giuseppe Faggian, Giunai Sefieva, Ilia Derevitskii, Anna Malashicheva, Evgeny Shlyakhto

**Affiliations:** 1Almazov National Medical Research Centre, 197341 Saint Petersburg, Russia; georgy.kopanitsa@gmail.com (G.K.); akostareva@hotmail.com (A.K.); olegmetsker@gmail.com (O.M.); vladimiruspenskiy@gmail.com (V.U.); mlgordeev@mail.ru (G.M.); gunaj4143@icloud.com (G.S.); amalashicheva@gmail.com (A.M.); e.shlyakhto@almazovcentre.ru (E.S.); 2Department of Cardiac Surgery, Medical School, ITMO University, 49 Kronverskiy Prospect, 197101 Saint Petersburg, Russia; ivderevitckii@itmo.ru; 3Department of Cardiac Surgery, Medical School, University of Verona, 37126 Verona, Italy; giuseppe.faggian@univr.it

**Keywords:** ascending aortic dilatation, aneurysm, risk factors, echocardiography

## Abstract

Aortic aneurysm (AA) rapture is one of the leading causes of death worldwide. Unfortunately, the diagnosis of AA is often verified after the onset of complications, in most cases after aortic rupture. The aim of this study was to evaluate the frequency of ascending aortic aneurysm (AscAA) and aortic dilatation (AD) in patients with cardiovascular diseases undergoing echocardiography, and to identify the main risk factors depending on the morphology of the aortic valve. We processed 84,851 echocardiographic (ECHO) records of 13,050 patients with aortic dilatation (AD) in the Almazov National Medical Research Centre from 2010 to 2018, using machine learning methodologies. Despite a high prevalence of AD, the main reason for the performed ECHO was coronary artery disease (CAD) and hypertension (HP) in 33.5% and 14.2% of the patient groups, respectively. The prevalence of ascending AD (>40 mm) was 15.4% (13,050 patients; 78.3% (10,212 patients) in men and 21.7% (2838 patients) in women). Only 1.6% (*n* = 212) of the 13,050 patients with AD knew about AD before undergoing ECHO in our center. Among all the patients who underwent ECHO, we identified 1544 (1.8%) with bicuspid aortic valve (BAV) and 635 with BAV had AD (only 4.8% of all AD patients). According to the results of the random forest feature importance analysis, we identified the eight main factors of AD: age, male sex, vmax aortic valve (AV), aortic stenosis (AS), blood pressure, aortic regurgitation (AR), diabetes mellitus, and heart failure (HF). The known factors of AD-like HP, CAD, hyperlipidemia, BAV, and obesity, were also AD risk factors, but were not as important. Our study showed a high frequency of AscAA and dilation. Standard risk factors of AscAA such as HP, hyperlipidemia, or obesity are significantly more common in patients with AD, but the main factors in the formation of AD are age, male sex, vmax AV, blood pressure, AS, AR, HF, and diabetes mellitus. In males with BAV, AD incidence did not differ significantly, but the presence of congenital heart disease was one of the 12 main risk factors for the formation of AD and association with more significant aortic dilatation in AscAA groups.

## 1. Introduction

According to different systematic reviews, the incidence of thoracic aortic aneurysms (TAA) in the general population is increasing in frequency from 5 to 10.4 per 100,000 patients [1,2,3]. However, there are no recommendations for screening for thoracic aortic aneurysms (TAAs) [4]. Only few studies have illustrated the role of different risk factors in the onset and progression of ascending aortic dilatation [5,6] The majority of them have shown that the prevalence of ascending aortic aneurysm increases with age and depends on sex and body surface area. It is also known that occurrences of ascending aorta aneurysm in males is twice higher than in females. Other recognized risk factors were arterial hypertension, atherosclerosis, tobacco smoking, dyslipidemia, and diabetes mellitus [2,5,6,7]. Dyslipidemia is a weaker risk factor, whereas diabetes generally reduces the risk of abdominal aortic aneurysm [8]. Furthermore, it has been shown that antihypertensive therapy and smoking cessation can modify the dimension of the aorta in patients with abdominal aortic aneurysms [9,10,11]. Only family history of AA, increase of the ascending aortic diameter over 3 mm per year, aortic coarctation and history of arterial hypertension are powerful predictors of aortic aneurysm [12]. On the other hand, it is known that BAV is currently one of the most common congenital heart defects. Its detectability according to various data in the general population varies from 0.5% to 2% of cases also associated with aortic pathology [13,14].

Recent ESC guidelines on the diagnosis and treatment of aortic diseases recommended beta blockers and ACE inhibitors to control hypertension, but still no pharmacological treatment is available to effectively reduce ascending aortic dilatation. Currently, open-heart surgery is the only effective treatment for patients with ascending aortic aneurysms and the main measurement to recommend intervention for aortic aneurysms for surgeons is aortic diameter [9]. It is the best predictor for operation, but there are many articles which demonstrate that complications occur frequently with small aortic sizes and the cause of this is unknown [15,16].

Furthermore, the progression of ascending aortic dilation to aortic dissection is well known. Incidence of aortic dissection is still estimated to range between 2.49 and 2.78 cases per 100,000 persons a year, and 85% of all cases are still undiagnosed before death [12].

Therefore, estimation of the individual risk profile, early diagnosis of ascending aortic dilatation (AscAD) in high-risk patients, and elective surgery are crucial to prevent AD and its potential progression to aortic dissection, rupture, and sudden death.

Over the years, medical centers have accumulated a significant amount of information about patient observations [17]. This information can be used for the statistical analysis and identification of characteristics of patients with diseases. This can be especially applicable for asymptomatic diseases. A significant number of disparate changes in indicators allows to identify the progression of diseases in the early stages.

This study describes clinical features and pathology patterns of a population of Russian patients with AscAA with the aim of identifying risk factors for early diagnosis of “silent” AscAD and aneurysm using machine learning methods, and characterizing their association with valve morphology and patient characteristics.

## 2. Materials and Methods

The study protocol was approved by the local ethics committee at the Almazov National Medical Research Centre in Saint Petersburg, Russian Federation, before the initiation of the study according to the principles of the Declaration of Helsinki.

### 2.1. Study Cohort

We retrospectively analyzed the ECHO database in the Almazov National Medical Research Centre to identify patients with aneurysms.

Furthermore, original methods of natural language processing were applied to extract the characteristics of BAV and tricuspid aortic valve (TAV) from electronic medical records including anamnesis, epicrisis, and results of instrumental tests [18,19].

This database included results of 145,454 ECHOs of outpatients and hospitalized patients, who were observed and treated in the centre between January 2010 and November 2018.

We used the following criteria to include and exclude patients in the study:

Inclusion criteria:Patient from the ECHO database whose treatment started after the 1st of January 2010 and ended before the 30th of November 2018;Diameter of the ascending aorta > 40 mm;For patients who underwent ECHO examination more than once during this period, only the first results of verified AD were included in the study. ECHO was most commonly performed in the following clinical situations: suspected cardiac etiology based on symptoms, signs or other testing; evaluation and follow-up of subjects with cardiovascular disease;Age ≥ 18 years old.

Exclusion criteria:5.Patients whose treatment started before the 1st of January 2010 or ended after the 30th of November 2018;6.Patients who did not have a complete data set.

The data set contained 50 predictors and 1 function with the following values: 1 for the patients with AD and 0 for the patients with no AD. A total of 84,851 cases that met the inclusion and exclusion criteria were analyzed retrospectively.

Detailed information including demographic characteristics, characteristics obtained through ECHO, and comorbidities were extracted from outpatient clinic physical exams, as well as from hospital charts related to hospitalizations occurring within a year before index echocardiography was performed. Comorbid diseases were similarly extracted from outpatient clinic and/or hospital admissions.

All patients were divided into 2 subgroups: patients with TAV, and patients with BAV. Demographic characteristics of all patients are presented in Table 1, subgroups in Table 2.

### 2.2. Echocardiography

All patients underwent comprehensive 2-dimensional and Doppler transthoracic echocardiography according to current echocardiography guidelines, using the Vivid 7.0 system (GE, Philadelphia, PA, USA) [10,11,12]. Aortic root planimetry (including diameters of the ascending aorta at different levels) was comprehensively assessed. Internal diameter was measured perpendicular to the axis of blood flow routinely obtained through the parasternal long-axis view.

Measurements of aortic diameters, ventricular sizes and function, and valve performance were conducted according to current recommendations on echocardiography [18,20,21]. Absolute value of the maximal aortic diameter was indexed to body surface area [22]. Diagnosis of BAV was based on short-axis imaging of the aortic valve, demonstrating the existence of only 2 commissures, delimiting only 2 aortic valve cusps. For each ECHO case, we also analyzed the reason why ECHO was ordered. ECHO was performed in patients with coronary artery disease (CAD) and hypertension in 33.5% and 14.2% of cases, respectively.

### 2.3. Statistical Methods

Statistical analysis was performed using STATISTICA v. 10.0 (StatSoft Inc., Tulsa, OK, USA). Baseline characteristics of the study population are given as percentages for qualitative variables, and medians and quartiles for quantitative variables (not normally distributed), as appropriate, by sex and aortic dilatation status. The *p*-test was applied to obtain the probability for the distribution of characteristic values of different groups of patients with BAV aneurysms compared to observed patients with TAV. Because of the demonstration of significant differences between sexes in terms of demographic characteristics, all the analyses were separately reported for men and women.

### 2.4. Data Preprocessing

We removed 1% of values having the highest z-score to filter out some obvious outliers. Furthermore, we applied the min and max normalization to the remaining values.

### 2.5. Classification Model and Feature Importance

Each experiment ran in the setting of stratified 5-fold cross-validation (i.e., randomly 80% of patients were used for training and 20% for testing, target class ratios in the folds were preserved). A random forest algorithm was applied to calculate the feature importance. The algorithm was implemented using Python 3.6.3 and the scikit-learn 0.19.1 (https://scikit-learn.org/stable/ (accessed on 12 May 2022)) library. A random forest (RF) is an ensemble of machine-learning algorithms, which is best defined as a “combination of tree predictors such that each tree depends on the values of a random vector sampled independently and with the same distribution for all trees in the forest”.

As an additional performance assessment score, we used the area under the curve (AUC) of the receiver operating characteristic (ROC), which represents the trade-off between sensitivity and specificity of the model. The AUC was calculated based on an average of 5 curves (one curve per fold in the setting of 5-fold cross-validation). All the measurements were performed separately per dataset and per model parameter value to determine the best parameters for classifiers as well as optimal data preprocessing. The hyperparameters optimization was performed and the results were obtained based on having hyperparameters tuned.

The *p*-value was calculated using the following methods: for each sample of the dead (<1, 1–3, 4–10), the *p*-value of the corresponding test was calculated for each column. Chi-square criterion was applied for categorical features, and Kolmogorov–Smirnov test was deployed for continuous features.

The experiments were conducted with Python 3 packages: scikit-learn [23] and Catboost [24] for machine-learning models implementation, seaborn [25] and matplotlib [26] for data visualization, smote [27] for dataset balancing, and SHapley Additive exPlanations (SHAP) [28] for the black-box results interpretation. The discrimination was evaluated using ROC curves. 

Table 3 lists the machine learning models and parameters used in the research.

## 3. Results

The population size of the study was 84,851 patients screened by ECHO.

The main reason to apply ECHO were CAD, suspected hypertension or known valvular heart disease (VHD), different variants of arrhythmia, and other reasons (Figure 1). Only 212 (0.25%) of patients had been aware of their aortic aneurysm before ECHO was performed in the Centre. 

However, 13,050 (15.4%) patients undergoing ECHO were diagnosed with aortic dilatations. Demographic and clinical characteristics of all patients, who underwent ECHO, are outlined in Table 1. The average age of all included patients was 55.8 (34:68) years. HF, CAD, HP, and obesity were more common in the cohort than dyslipidemia or VHD. Average EF of LV, BP, were normal. The main characteristics of 13,050 patients with aortic dilation are shown in Table 2. Among them, the majority were male (10,202 patients, representing 78.2%). Table 4 and Table 5 present demographic and clinical characteristics of all the patients.

Feature importance: According to the analysis of the significance of predictors by the random forest method, the greatest contribution to the development of aortic aneurysm is made by age and male sex (cut off = 0.25). Less significant contributions were provided by Vmax AV, AS, blood pressure, AR, HF (cut off = 0.025), and other factors such as diabetes mellitus, HP, CAD, fibrillation, asthma, obstructive pulmonary disease (COPD), hyperlipidemia, stroke, thyroid disorders, cholecystitis, congenital heart disease (CHD), BAV, and obesity (Figure 2).

AUC of ROC was calculated with a value of 0.92. The resulting ROC curve is presented in Figure 3.

Among all patients who underwent ECHO, we identified 1544 (1.8%) with BAV, and 645 patients with BAV had AD (only 4.9% of all patients with AD).

According to the results of the analysis of all patients who underwent ECHO dependent on valve morphology, a more significant aortic dilatation was observed in the group with BAV (Table 2). Besides, AS and AR verification happened in BAV patients four times more often compared to patients without CHD. The latter were younger and had HF more frequently (*p* < 0.0001), while HP, CAD, obesity, hyperlipidemia, and diabetes mellitus were more frequently registered in patients with TAV (*p* < 0.0001). 

All patients with AD (Table 3), regardless of valve morphology, were older, and rates of obesity, hyperlipidemia, and HF were higher (*p* < 0.003).

Aortic regurgitation (AR) is more often diagnosed in TAV patients with AD (Figure 4). Aortic stenosis (AS) was frequently verified in all AD patients besides women with BAV (Figure 4). Furthermore, in the group of men with BAV and AD, CAD and COPD were more frequently registered than in patients without AD (Table 3) (*p* < 0.0001).

In the TAV group with AD rates of CAD, asthma (Table 3) was more frequent than in the group without AD regardless of sex (*p* < 0.02). Besides, patients from the TAV group with AD had higher blood pressure (BP) than patients without AD, also regardless of sex (*p* < 0.0001). Hypotensive and lipid-reducing therapy was more common in patients with aortic aneurysm according to our register. Table 3 shows the best performances for each classification target.

## 4. Discussion

The present study demonstrates that the prevalence of aortic ascending dilatation is 15.4% over 84,851 individuals in our region, a proportion higher than the average incidence of thoracic aortic aneurysm (5–10.4 per 100,000 population) reported in the general population [19]. Compared with epidemiological studies in other countries, our study showed a relatively higher prevalence. This discrepancy of risk in ascending aneurysms compared with the general population might be explained in part by the nature of individuals in the current study. The study subjects were not selected incidentally from the general population, but they initially underwent elective TTE due to a cardiologic clinical indication. We compared our data with another retrospective study, which was also based on a medical center’s database. Wang et al. showed that the average incidence among the elderly was 56.1 per 100,000 [29,30]. Unfortunately, our study showed that only 1.6% (n = 212) of 13,050 patients with AD diagnosed by ECHO were aware of their disease. Indeed, in the majority of cases, the disease was accidentally diagnosed during examinations of other diseases, given that the main indication for echocardiography examination was CAD and HP.

The previous study as well as our analysis revealed that women are less likely to have aortic aneurysms compared to men, but women with abdominal aortic aneurysms are at higher risks of aneurysmal dissection, rupture, and complications-related mortality than men [17,18].

In our study, we identified significant risk factors associated with ascending aortic dilatation. According to our results, patients with AD were older than patients without AD, suffered more often from obesity, and had hyperlipidemia and HF more often. However, only males with AD in BAV patients have more often shown CAD, COPD, and AS. Besides, TAV patients with AD were older than TAV patients without AD, had higher BP, suffered more often from obesity, and had hyperlipidemia, CAD, HF, AR, and AS more often. Low prevalence of CAD, HP, and COPD in patients with BAV can reflect another mechanism of the formation and a possible influence of genetic factors [31].

We also observed a higher prevalence of aortic regurgitation and AS in all groups with AD besides the frequency of AS in females with BAV. During the analysis, both men and women had an AS and AR frequency four times higher in patients with BAV.

However, given that the number of females with BAV and AD is less than that of other patients, this did not affect the results of the machine learning analysis of the significance of AD development predictors. According to the results of the machine learning analysis, it is AS that has the greatest impact on the development of AD. On the one hand, the data we obtained contradicts the analysis conducted by Boudoulas et al., which demonstrated that aortic pathology in combination with AS was mostly found in patients with BAV, while only 3% of cases were found in patients with TAV [32].

On the other hand, the results of this analysis once again confirm the contribution of the hemodynamic role of AD formation called “post-stenotic aortic dilatation” due to the influence of the high-velocity turbulent transaortic jet on the aortic wall. An interesting fact is the contribution of diabetes mellitus, CAD, asthma, and COPD to the formation of AD. In general, the understanding of the formation of these diseases follows an inflammatory theory. In particular, Liu. et al. demonstrated a possible mechanism for increasing the size of the abdominal aortic aneurysm in the presence of a lung asthma allergic disease through the activation of an immuno-inflammatory pathway in mice. Intraperitoneal administration of an anti-IgE antibody suppressed AAA lesion formation and reduced lesion inflammation, plasma IgE, and bronchioalveolar inflammation [33]. Rosa et al. showed an inverse association of soluble IL6R with abdominal aortic aneurysm [34]. As for diabetes mellitus, there is currently conflicting data from some authors about its protective effect on patients with AD [35,36,37]. The results of other studies confirm the negative role of diabetes mellitus for patients with AD [38,39]. Furthermore, Ortega et al. demonstrated that inhibition of SGLT-2 by empagliflozin inhibits AAA formation and can represent a promising therapeutic strategy to prevent AAA progression [40].

In conclusion, our data confirms the importance of echocardiography screening in patients with a known risk factor for aortic dilation. In addition to patients with a known diagnosis of AS, AR, and CAD, ECHO should be performed in patients with diabetes mellitus and asthma, especially in older males. Similarly, if BAV was diagnosed, but aortic dilatation or another pathology was not verified, it is necessary to repeat a control echocardiography, especially if age exceeds 40 years. For example, the risk of aneurysm is 26% at 25 years after the initial diagnosis of BAV at echocardiography, with an incidence of approximately 85 cases per 10 000 patient–years, which represents 80 times the risk of aneurysm formation of the general population [41]. However, the follow-up time interval between BAV diagnosis and aneurysm formation is unpredictable in a single patient. Therefore, a strong recommendation for annual follow-up shall be made for ascending aortic diameters > 45 mm. Large clinical registries based on electronic health records (EHRs) may be used to assess different treatment strategies, to evaluate multiple risk factors and/or outcomes simultaneously, to test associations in subpopulations, to analyze longitudinal outcomes and adverse effects for large cohorts of diverse patients, and to capture uncommon diseases or conditions that are rarely examined in traditional clinical trials. However, analyzing these data is not easy due to differences in EHR encoding systems and data fragmentation across practices and institutions.

### Study Limitation

This study is based on a retrospective analysis of clinically obtained patient data derived from a single center. It is not truly a population-based research. All patients were admitted and followed-up in a tertiary referral center. This may have led to a selection bias of a higher risky population of patients. Both biases can be minimized by the inclusion of all consecutive patients undergoing echocardiography between 2010 and 2018. Another limitation of this study was the selection of patients based only on the performance of an echocardiographic study.

## 5. Conclusions

Our study showed a high prevalence of ascending aortic dilatation, most of which is completely asymptomatic. This fact puts these patients at great risk of severe aortic-related complications. Recognition of risk factors for aortic dilatation can lead to individualized screening programs. AS, AR, diabetes mellitus, CAD, and asthma are established major risk factors for echocardiography screening in order to identify unknown ascending aorta aneurysms, especially if the patient has other risk factors for AD such as COPD, obesity, hyperlipidemia, HP, BAV, age, and male sex.

The need for such ultrasound screening in patients with ascending aorta aneurysm is confirmed by the results of a similar survey of patients with abdominal aneurysm, which demonstrated a reduction in aneurysm-related mortality in men older than 65 years [42,43].

We have identified the main risk factors, the commonality of which results in the activation of the immuno-inflammatory system, which can help in finding therapeutic targets for the treatment of thoracic aortic dilation.

## Figures and Tables

**Figure 1 jpm-12-00794-f001:**
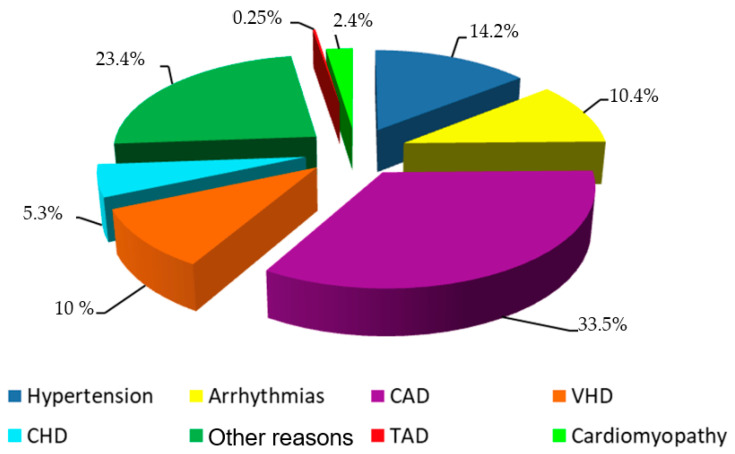
The pipeline for medical risk model development.

**Figure 2 jpm-12-00794-f002:**
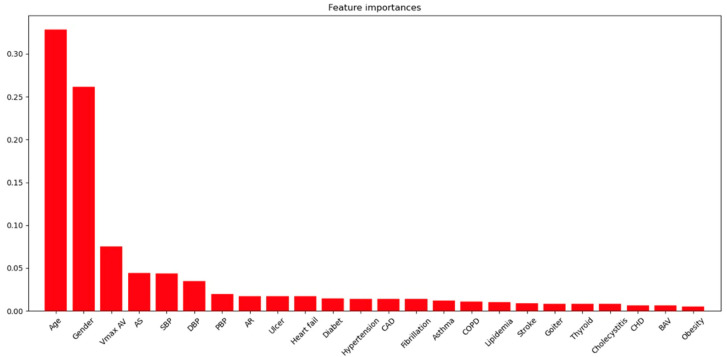
Features importance analysis.

**Figure 3 jpm-12-00794-f003:**
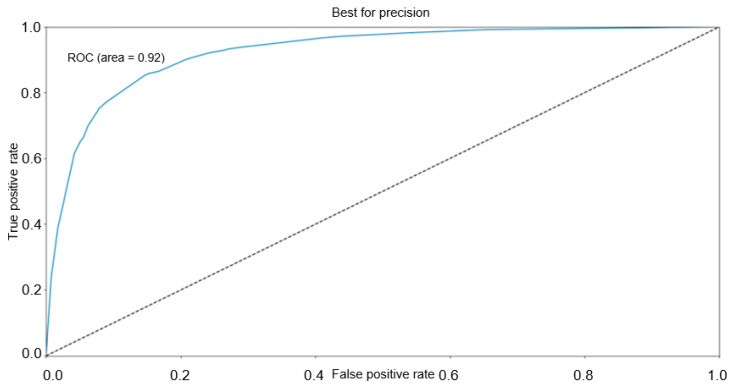
ROC for the classification model.

**Figure 4 jpm-12-00794-f004:**
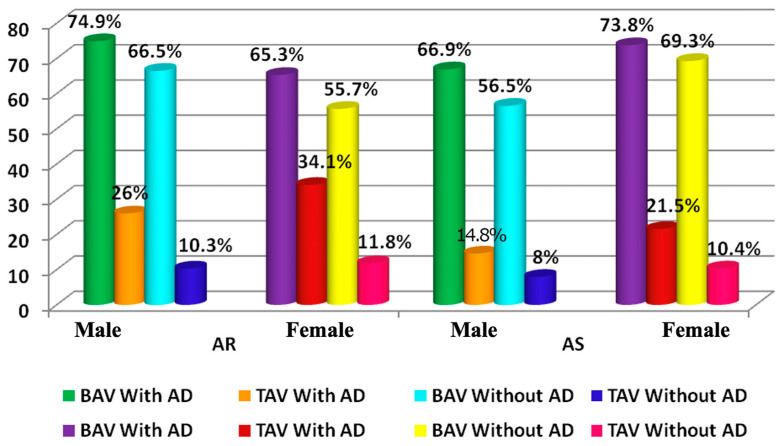
The frequency of the AR and AS in patients with/without AD and BAV; AR—aortic regurgitation; AS—aortic stenosis.

**Table 1 jpm-12-00794-t001:** Demographic and clinical characteristics of all patients.

Variables	N	All Patients	Min/Max
Age, years (median; quartiles)	84,851	59 (34; 68)	18; 107
Aortic diameter at the sinus of the Valsalva, mm, median; quartiles	84,851	34 (31; 37)	8; 90
Aortic diameter at the proximal ascending aorta, mm, median; quartiles	84,851	33 (30; 36)	12; 98
BMI, kg/m^2^, median; quartiles	27,362	27.3 (21.4; 31.0)	12.5; 97.7
AS dpmax, mmHg, median; quartiles	84,851	7.0 (5.0; 10.0)	0.36; 424
EF LV (%), median; quartiles	76,800	63.9 (56.9; 68.9)	7.0; 91.5
SBP office, mmHg, median; quartiles	84,851	130 (120; 142)	55; 270
DBP office, mmHg, median; quartiles	84,851	80 (80; 87)	20; 140
AR, *n* (%)	84,757	4460 (5.26)	-
AS, *n* (%)	84,851	11,252 (13.26)	-
Diabetes mellitus, *n* (%)	84,851	8426 (9.93)	-
Hypertension, *n* (%)	84,851	59,711 (70.37)	-
CAD, *n* (%)	84,851	28,440 (33.52)	-
COPD, *n* (%)	84,851	6818 (8.04)	-
Asthma, *n* (%)	84,851	2207 (2.60)	-
Obesity, (BMI > 30), *n* (%)	27,362	8420 (30.77)	-
Hyperlipidemia, *n* (%)	84,851	21,087(24.85)	-
Heart failure, *n* (%)	84,851	35,194 (41.48)	-

BMI—body mass index; SBP—systolic blood pressure; DBP—diastolic blood pressure; AS dpmax—antegrade gradient across the narrowed aortic valve; AR—aortic regurgitation; AS—aortic stenosis; COPD—chronic obstructive pulmonary disease; CAD—coronary artery disease.

**Table 2 jpm-12-00794-t002:** Demographic and clinical characteristics of all patients depend on valve morphology.

Variables	BAV, *n* = 1544 N; Median; Quartiles	TAV, *n* = 83,317 N; Median; Quartiles	*p*
Age, years (median and bounds)	40.5 (18; 104)	59 (18; 88)	<0.0001
Aortic diameter at the sinus of the Valsalva, mm	35 (32; 39)	34 (31; 37)	<0.0001
Aortic diameter at the proximal ascending aorta, mm	36 (32; 42)	33 (30; 36)	<0.0001
BMI, kg/m^2^	25.5 (22.8; 28.6)	27.3 (24.2; 31.1)	<0.0001
AS dpmax, mmHg	18 (11; 34)	7 (5; 10)	<0.0001
EF LV (%), ΦB	65.0 (59.7; 70.1)	63.9 (56.9; 68.9)	<0.0001
SBP office, mmHg	130 (120; 140)	130 (120; 143)	0.008
DBP office, mmHg	80 (73.5; 83.5)	80 (80; 88)	0.0006
AR, *n* (%)	333 (21.72)	4127 (4.96)	<0.0001
AS, *n* (%)	901 (58.77)	10,351 (12.42)	<0.0001
Diabetes mellitus, *n* (%)	77 (5.02)	8349 (10.02)	<0.0001
Hypertension, *n* (%)	861 (56.13)	58,850 (70.63)	<0.0001
CAD, *n* (%)	249 (16.23)	28,191 (33.84)	<0.0001
COPD, *n* (%)	101 (6.58)	6717 (8.06)	0.03
Asthma, *n* (%)	46 (3.00)	2161 (2.59)	0.32
Obesity, (BMI > 30), *n* (%)	133 (18.68)	8287 (31.10)	<0.0001
Hyperlipidemia, *n* (%)	305 (19.88)	20,782 (24.94)	<0.0001
Heart failure, *n* (%)	750 (48.89)	34,444 (41.34)	<0.0001

BMI—body mass index; SBP—systolic blood pressure; DBP—diastolic blood pressure; AS dpmax—antegrade gradient across the narrowed aortic valve; AR—aortic regurgitation; AS—aortic stenosis; COPD—chronic obstructive pulmonary disease; CAD—coronary artery disease.

**Table 3 jpm-12-00794-t003:** Models and parameters.

Model	Parameters
LR * (imp. feat.)	‘C’: 2.83, ‘solver’: ‘newton-cg’
LR + SMOTE (imp. feat.)	‘C’: 0.5, ‘solver’: ‘newton-cg’
LR + SMOTE (all feat.)	‘C’: 4.0, ‘solver’: ‘liblinear’
RF (imp. feat.)	‘criterion’: ‘gini’, ‘max_features’: ‘auto’
RF + SMOTE (imp. feat.)	‘criterion’: ‘gini’, ‘max_features’: ‘auto’
RF + SMOTE (all feat.)	‘criterion’: ‘gini’, ‘max_features’: ‘log2’
CC * (all. feat.)	‘depth’: 4, ‘l2_leaf_reg’: 3, ‘learning_rate’: 0.6
CC + SMOTE (imp. feat.)	‘depth’: 5, ‘l2_leaf_reg’: 2, ‘learning_rate’: 0.9
CC + SMOTE (all feat.)	‘depth’: 4, ‘l2_leaf_reg’: 1, ‘learning_rate’: 0.2

* LR—logistic regression; SMOTE—Synthetic Minority Oversampling Technique; RF—random forest; CC—catboost classifier; imp. feat.—the model is composed using only important features; all feat.—the model is composed using all available features.

**Table 4 jpm-12-00794-t004:** Demographic and clinical characteristics of female patients.

Variables	BAV		TAV	
	With AD, *n* = 150	Without AD, *n* = 400	With AD, *n* = 2688	Without AD, *n* = 40,925
Age, years	54.5	35	67	59
Aortic diameter (sinus Valsalva)	36	31	38	32
Aortic diameter proximal ascending aorta	44 (41; 47)	33 (30; 4)	41 (40; 44)	31 (29; 34)
BMI, kg/m^2^	26.7	24.5	28.7	27.2
ASdpmax, mm Hg.	22 (14; 44)	22 (12; 42.5)	10 (6; 19)	7 (6; 10)
EF LV (%),	66.2	66.9	64.2	65.8
SBP office, mm Hg.	122.5	120	135	130
DBPoffice, mm Hg.	80	80	80	80
AR, *n* (%)	24 (16.0)	67 (16.75)	372 (13.9)	1691 (4.1)
AS, *n* (%)	105 (70.0)	260 (65.0)	824 (30.7)	5104 (12.5)
Diabetes mellitus, *n* (%)	8 (5.33)	21 (5.25)	305 (11.4)	4389 (10.7)
CAD, *n* (%)	25 (16.67)	51 (12.75)	1041 (38.7)	11,110 (27.2)
COPD, *n* (%)	5 (3.33)	15 (3.75)	205 (7.6)	2365 (5.8)
Asthma, *n* (%)	5 (3.33)	10 (2.50)	122 (4.5)	1219 (2.9)
Obesity, *n* (%)	24 (30.77)	28 (13.93)	392 (42.2)	4496 (32.2)
Hyperlipidemia, *n* (%)	51 (34.0)	62 (15.50)	856 (31.9)	9695 (23.7)
Heart failure, *n* (%)	26.7 (24.5; 32.1)	24.5 (21.5; 27.5)	28.7 (25.1; 32.9)	27.2 (23.7; 31.2)

**Table 5 jpm-12-00794-t005:** Demographic and clinical characteristics of male patients.

Variables	BAV		TAV	
	With AD, *n* = 495	Without AD, *n* = 499	With AD, *n* = 9717	Without AD, *n* = 29,987
Age, years,	50	29	63	56
Aortic diameter (sinus Valsalva)	41	34	41	35
Aortic diameter proximal ascending aorta	42 (40; 46)	33 (30; 36)	40 (37; 42)	33 (31; 36)
BMI, kg/m^2^,	27.1	24.3	28.4	27.0
ASdpmax, mm Hg.	20 (12; 39)	14,5 (9; 26)	7 (5; 11)	6 (5; 8)
EF LV (%),	62.9	64.9	59.4	61.6
SBP office, mm Hg.	135	130	130	130
DBPoffice, mm Hg.	80	80	80	80
AR, *n* (%)	129 (26.6)	113 (22.7)	1029 (10.6)	1035(3.4)
AS, *n* (%)	302 (62.3)	234 (46.9)	1661 (17.1)	2762 (9.2)
Diabetes mellitus, *n* (%)	24 (4.95)	24 (4.8)	953 (9.8)	2702 (9.0)
CAD, *n* (%)	119 (24.54)	54 (10.8)	4454 (45.8)	11,586 (38.6)
COPD, *n* (%)	52 (10.7)	29 (5.8)	1045 (10.8)	3102 (10.3)
Asthma, *n* (%)	16 (3.3)	15 (3.0)	230 (2.4)	590 (1.9)
Obesity, *n* (%)	53 (24.31)	28 (13.0)	1046 (36.9)	2353 (26.5)
Hyperlipidemia, *n* (%)	122 (25.2)	70 (14.0)	2901 (29.9)	7330 (24.4)
Heart failure, *n* (%)	282 (58.1)	195 (39.1)	4854 (49.9)	12248 (40.8)

## Data Availability

Not applicable.

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
