# Peer review of "Application of Machine Learning Methods to Analyze Occurrence and Clinical Features of Ascending Aortic Dilatation in Patients with and without Bicuspid Aortic Valve"

_jpm, 2022, doi:10.3390/jpm12050794_

Round 1
Reviewer 1 Report
Please provide answers to the following points and incorporate in the paper. The research is of high quality, but there are some points that needs to be answered before final acceptance.
Q1
Authors found 8 factors as very important in respect to aortic dilatation (AD): age, male gender, vmax aortic valve (AV), aortic stenosis (AS), blood pressure, aortic regurgitation (AR), diabetes mellitus, and heart failure (HF) and some factors, such as: AD-like HP, CAD, hyperlipidemia, BAV, and obesity as less important.
Please state the cut-off point or the decision threshold upon what you have selected some factors as more and others as less important ?
Q2
Please provide reference (link) to ECHO database if available (include at the beginning of Section 2, when you first mentioned ECHO database)?
Q3
Abbreviations under Table 2 should be placed at the beginning of the paper, following abstract and keywords, providing full names of all abbreviations used in the paper, computational and medical.
Q4
You have applied Machine Learning, having hyperparameters tuned. That’s not a problem, but you must state in the paper the hyperparameters optimization was performed and the results are obtained based on having hyperparameters tuned, that in the most cases will generate near-optimal or optimal results.
Q5
You have stated that you have performed stratified cross-validation and calculated AUC ROC. I did not see any report of the findings for AUC ROC reported in the paper. Please provide the AUC results that you got, as well as figures for ROC curve(s) ?
Author Response
Dear reviewer 1, please see the answers to your comments:
Q1
Authors found 8 factors as very important in respect to aortic dilatation (AD): age, male gender, vmax aortic valve (AV), aortic stenosis (AS), blood pressure, aortic regurgitation (AR), diabetes mellitus, and heart failure (HF) and some factors, such as: AD-like HP, CAD, hyperlipidemia, BAV, and obesity as less important.
Please state the cut-off point or the decision threshold upon what you have selected some factors as more and others as less important ?
Feature importance: According to the analysis of the significance of predictors by the random forest method, the greatest contribution to the development of aortic aneurysm is made by age and male gender (Cut off = 0.25). Less significant contributions were provided by Vmax AV, AS, blood pressure, AR, HF (Cut off = 0.025), and other factors like diabetes mellitus, HP, CAD, fibrillation, asthma, obstructive pulmonary disease (COPD), hyperlipidemia, stroke, thyroid disorders, cholecystitis, congenital heart disease (CHD), BAV, and obesity (Figure 2).Q2
Please provide reference (link) to ECHO database if available (include at the beginning of Section 2, when you first mentioned ECHO database)?
The ECHO database is not available publicly now. It is a part of the Almazov National Research Medical Center infrastructure.
Q3
Abbreviations under Table 2 should be placed at the beginning of the paper, following abstract and keywords, providing full names of all abbreviations used in the paper, computational and medical.
The abbreviations were placed in the beginning of the paper.
Abbreviations
- AA – Aortic aneurysm
- AscAA – ascending aortic aneurysm
- AS – dpmax – antegrade gradient across the narrowed aortic valve,
- AD – aortic dilatation
- AR – aortic regurgitation,
- AS – aortic stenosis,
- AV – aortic valve
- BAV – bicuspid aortic valve
- BMI – body mass index,
- CAD – Coronary Artery Disease
- CHD – congenital heart disease
- COPD – chronic obstructive pulmonary disease,
- DBP – diastolic blood pressure,
- ECHO – echocardiographic
- HF – heart failure
- HP – hypertension
- SBP – systolic blood pressure,
- TAA – thoracic aortic aneurysms
- VHD – valvular heart disease
Q4
You have applied Machine Learning, having hyperparameters tuned. That’s not a problem, but you must state in the paper the hyperparameters optimization was performed and the results are obtained based on having hyperparameters tuned, that in the most cases will generate near-optimal or optimal results.
A statement was added to the methods section:
The hyperparameters optimization was performed and the results were obtained based on having hyperparameters tuned.
Q5
You have stated that you have performed stratified cross-validation and calculated AUC ROC. I did not see any report of the findings for AUC ROC reported in the paper. Please provide the AUC results that you got, as well as figures for ROC curve(s) ?
AUC of ROC was calculated with a value of 0,92. The resulting ROC curve is presented in the figure 3.
Figure 3. ROC for the classification model
Reviewer 2 Report
It is evident that the paper is focused on the application of data mining techniques in the analysis of data collected in routine work, recorded in the ECHO database and linked with patient's data (EMR). My question is: How are the records in the ECHO database linked to corresponding EMRs? Patient identification? Anything else?
The problem of identifying risk factors for early diagnosis of "silent" ascending aortic dilatation is important issue "to prevent AD and its potential progression...". Are there exclusively thoracic aortic aneurysms included or abdominal ones too?
Inclusion and exclusion criteria for patients from ECHO database are clearly described, as well as statistical methods. In Classification model and feature importance there is Table 2 (? - seems it should be Table 3) with models and parameters. Table and its description is difficult for understanding. In such form it does not add any value to this paper. So, make it more useful.
In Results: Read carefully and pay attention to the numbering of the tables. Table 3 (? - seems it should be Table 4) is completely not transparent - divide it into two tables. You found that AR and AS are important features (variables) from echocardiography data identified as risks for early diagnosis of "silent" ascending aortic dilatation is important issue "to prevent AD and its potential progression...". How to use this knowledge in practice (for individual or for population)? Is there any recommendation for practice based on your research? .Can you formulate conclusion coming from your research, or suggestions on how and what to do next?
One more comment: in Abstract - AA is NOT "one of the leading causes of death". Maybe mortality rate in case of rupture AA can be considered as such.
Author Response
Dear Reviewer 2, please see the answers to your comments:
It is evident that the paper is focused on the application of data mining techniques in the analysis of data collected in routine work, recorded in the ECHO database and linked with patient's data (EMR). My question is: How are the records in the ECHO database linked to corresponding EMRs? Patient identification? Anything else?
The informed consent was waived due to the anonymous nature of the data that was used in the study. This doesn’tallow to identify a patient. The records in the ECHO database were linked with the EMR data by a patient ID. This link was removed in the preprocessing phase before the study begin to prevent identification of the patients.
The problem of identifying risk factors for early diagnosis of "silent" ascending aortic dilatation is important issue "to prevent AD and its potential progression...". Are there exclusively thoracic aortic aneurysms included or abdominal ones too?
The study included only aortic aneurism. Abdominal aneurism will be a topic of the future work
Inclusion and exclusion criteria for patients from ECHO database are clearly described, as well as statistical methods. In Classification model and feature importance there is Table 2 (? - seems it should be Table 3) with models and parameters. Table and its description is difficult for understanding. In such form it does not add any value to this paper. So, make it more useful. In Results: Read carefully and pay attention to the numbering of the tables. Table 3 (? - seems it should be Table 4) is completely not transparent - divide it into two tables.
The tables’ numbering has been fixed. Now it is correct and consistent. We also split the tabvle 3 into 2 tables to make the results clearer.
You found that AR and AS are important features (variables) from echocardiography data identified as risks for early diagnosis of "silent" ascending aortic dilatation is important issue "to prevent AD and its potential progression...". How to use this knowledge in practice (for individual or for population)? Is there any recommendation for practice based on your research? .Can you formulate conclusion coming from your research, or suggestions on how and what to do next?
AR and AS were identified as important features from echocardiography data identified as risks that should be considered in the early diagnostics of "silent" ascending aortic dilatation to prevent AD and its potential progression. These findings require a validation in the follow up study. We think that a good tactics would be an additional CT control even in case of normal ECHO findings. These hypothesis requires a further investigation.
One more comment: in Abstract - AA is NOT "one of the leading causes of death". Maybe mortality rate in case of rupture AA can be considered as such.
It was corrected to: Aortic aneurysm (AA) rapture is one of the leading causes of death worldwide.